# Gut Microbiota Signatures in Colorectal Cancer as a Potential Diagnostic Biomarker in the Future: A Systematic Review

**DOI:** 10.3390/ijms25147937

**Published:** 2024-07-20

**Authors:** Lucian-Flavius Herlo, Andreea Salcudean, Roxana Sirli, Stela Iurciuc, Alexandra Herlo, Andreea Nelson-Twakor, Luana Alexandrescu, Raluca Dumache

**Affiliations:** 1Doctoral School, Victor Babes University of Medicine and Pharmacy, 300041 Timisoara, Romania; flavius.herlo@umft.ro; 2Discipline of Sociobiology, Department of Ethics and Social Sciences, George Emil Palade University of Medicine, Pharmacy, Science and Technology of Targu Mures, 540136 Targu Mures, Romania; andreea.salcudean@umfst.ro; 3Advanced Regional Research Center in Gastroenterology and Hepatology, Victor Babes University of Medicine and Pharmacy, 300041 Timisoara, Romania; sirli.roxana@umft.ro; 4Cardiology Department, Victor Babes University of Medicine and Pharmacy, 300041 Timisoara, Romania; iurciuc.stela@umft.ro; 5Department XIII, Discipline of Infectious Diseases, Victor Babes University of Medicine and Pharmacy Timisoara, 2 Eftimie Murgu Square, 300041 Timisoara, Romania; 6Department of Internal Medicine, County Clinical Emergency Hospital of Constanta, 900647 Constanta, Romania; andreeanelsont@gmail.com; 7Department of Gastroenterology, County Clinical Emergency Hospital of Constanta, 900647 Constanta, Romania; alexandrescu_l@yahoo.com; 8Department of Forensic Medicine, Bioethics, Medical ethics and Medical Law, Victor Babes University of Medicine and Pharmacy Timisoara, 300041 Timisoara, Romania; raluca.dumache@umft.ro; 9Center for Ethics in Human Genetic Identifications, Victor Babes University of Medicine and Pharmacy Timisoara, E. Murgu Square, Nr. 2, 300041 Timisoara, Romania

**Keywords:** colorectal cancer, gut microbiota, biomarker, microbiota composition

## Abstract

The gut microbiota has acquired significant attention in recent years for its potential as a diagnostic biomarker for colorectal cancer (CRC). In this literature review, we looked at the studies exploring alterations in gut microbiota composition associated with CRC, the potential mechanisms linking gut dysbiosis to CRC development, and the diagnostic approaches utilizing gut microbiota analysis. Our research has led to the conclusion that individuals with CRC often display alterations in their gut microbiota composition compared to healthy individuals. These alterations can include changes in the diversity, abundance, and type of bacteria present in the gut. While the use of gut microbiota as a diagnostic biomarker for CRC holds promise, further research is needed to validate its effectiveness and standardize testing protocols. Additionally, considerations such as variability in the microbiota composition among individuals and potential factors must be addressed before microbiota-based tests can be widely implemented in clinical practice.

## 1. Introduction

Colorectal cancer is a malignant tumor that develops in the colon or rectum, which are parts of the large intestine. It arises from the uncontrolled growth of abnormal cells in the lining of the colon or rectum and can spread to other parts of the body if not detected and treated early [1]. CRC is one of the most common cancers worldwide with significant morbidity and mortality [2].

CRC is the third most commonly diagnosed cancer in both men and women globally [3]. The incidence varies geographically with higher rates in developed countries compared to developing nations [4]. However, incidence rates are rising in many regions, including parts of Asia and Africa, due to changes in lifestyle, diet, and aging populations [5]. CRC is also the third leading cause of cancer-related deaths worldwide. Mortality rates have been declining in some high-income countries due to improved screening programs, early detection, and advances in treatment [6]. However, in low- and middle-income countries, where access to healthcare and screening is limited, mortality rates remain high [7].

Several risk factors contribute to the development of CRC, including age (risk increases with age), family history of CRC or polyps, inherited genetic syndromes (e.g., Lynch syndrome, familial adenomatous polyposis), personal history of inflammatory bowel disease (e.g., Crohn’s disease, ulcerative colitis), sedentary lifestyle, obesity, smoking, heavy alcohol consumption, and a diet high in red and processed meats and low in fruits, vegetables, and fiber [8].

Early detection of CRC through screening can significantly improve outcomes. Common screening methods include fecal occult blood tests, colonoscopy, sigmoidoscopy, and stool DNA tests [9]. Lifestyle modifications such as maintaining a healthy weight, regular physical activity, avoiding smoking and excessive alcohol consumption, and adopting a diet rich in fruits, vegetables, and whole grains can help reduce the risk of developing CRC [10].

### 1.1. Gut Microbiota

The gut microbiota is a complex community of trillions of microorganisms, including bacteria, viruses, fungi, and other single-celled organisms, which inhabit the gastrointestinal tract [11]. This ecosystem plays a crucial role in human health and disease through various mechanisms. It helps break down dietary components that the human digestive system cannot process alone, such as complex carbohydrates and fibers [12]. These microorganisms produce enzymes that facilitate the digestion of these substances, releasing nutrients that can then be absorbed by the host [13]. It also plays a vital role in regulating the immune system by helping the immune system distinguish between harmless substances and pathogens, thereby preventing inappropriate immune responses that can lead to inflammation or autoimmune diseases [14].

The microbiome contributes to maintaining the integrity of the intestinal barrier, which acts as a physical and immunological barrier between the gut lumen and the bloodstream [15]. A healthy gut microbiota helps prevent the translocation of harmful bacteria and toxins from the gut into systemic circulation, which can trigger inflammatory responses and contribute to various diseases [16]. Certain members are capable of synthesizing vitamins, such as vitamin K and some B vitamins, which are essential for human health. Additionally, gut microbes produce various metabolites, including short-chain fatty acids (SCFAs), which play important roles in energy metabolism, gut motility, and immune regulation [17].

Emerging research suggests a bidirectional communication system, known as the gut–brain axis, through which the gut microbiota can influence neurological and psychological function [18]. Alterations in the gut microbiota composition have been linked to conditions such as anxiety, depression, and neurodegenerative diseases [19].

Dysbiosis, or imbalance in the gut microbiota composition, has been implicated in the pathogenesis of numerous diseases, including inflammatory bowel diseases (e.g., Crohn’s disease, ulcerative colitis), obesity, type 2 diabetes, allergies, and certain cancers [20]. Understanding the role of the gut microbiota in these conditions may lead to the development of novel therapeutic approaches targeting the microbiome [21].

### 1.2. Gut Microbiota Dysbiosis in Colorectal Cancer

Gut microbiota dysbiosis, characterized by alterations in the composition and function of the gut microbiota, has been implicated in the development and progression of colorectal cancer [22].

Dysbiosis in the gut microbiota can lead to chronic inflammation and immune dysregulation within the intestinal mucosa, creating an environment conducive to CRC development. Certain bacterial species, such as Fusobacterium nucleatum and enterotoxigenic Bacteroides fragilis, have been associated with pro-inflammatory responses and promotion of CRC progression [23].

Dysbiosis in gut microbiota may produce metabolites that are carcinogenic or promote tumorigenesis. For example, some bacteria produce genotoxic metabolites such as hydrogen sulfide, secondary bile acids, and certain enzymes that can damage DNA and contribute to CRC initiation [24]. It can also compromise the integrity of the intestinal barrier, allowing the translocation of microbial products and toxins into the systemic circulation. This can trigger inflammation and immune responses both locally in the gut and systemically, which may contribute to CRC development and progression [25].

Metabolic dysregulation, such as obesity and insulin resistance, is a known risk factor for CRC. It interacts with the host immune system, affecting immune surveillance mechanisms against tumor cells [26]. Certain microbial species may modulate immune responses in a way that promotes tumor growth and the evasion of immune detection [27].

Understanding the role of gut microbiota dysbiosis in CRC development and progression is an active area of research. Therapeutic strategies aimed at restoring a healthy gut microbiota composition, such as probiotics, prebiotics, dietary interventions, and fecal microbiota transplantation, are being investigated as potential adjunctive treatments for CRC prevention and management [28].

## 2. Materials and Methods

We conducted a comprehensive literature search on PubMed using the specific keywords “microbiota and colorectal cancer”, “bacteria in colorectal cancer”, and “colorectal cancer and biomarkers”. We then performed a manual search of all eligible papers by using the references from the first search results, reviews, and other relevant publications. As this study is a literature review, an ethical approval was not needed.

The selection criteria was limited to free complete texts in English and randomized clinical trials involving people aged 18 and older. Only articles that were published throughout the timeframe of January 2013 to December 2023 were taken into account, and this review excluded articles that were restricted to abstracts, posters, editorials, and comments.

The exclusion criteria included age (under 18 years old), as well as non-peer-reviewed research. Case studies were also excluded. Studies without sufficient data and those without quantifiable results for outcomes were likewise removed. 

For the final studies that we included in our analysis, we used a systematic review technique using the Patient, Intervention, Comparison, Outcome (PICO) framework, as it was defined by Eriksen and Frandsen in 2018 [29].

Population: patients aged 18 and above diagnosed with CRC.

Intervention: methods of analysis of microbiota.

Comparison: gut microbiota of CRC patients in comparison with the healthy controls.

Outcome: to establish the relationship between microbiome and CRC.

The review was outlined here using the Preferred Reporting Items for Systematic Review and Meta-Analysis (PRISMA) guidelines (Figure 1).

A total of 698 citations were retrieved after scanning the aforementioned databases. After eliminating duplicate entries and excluding 113 items that did not satisfy the search parameters, the list was reduced to 127 remaining articles. A total of 64 studies were excluded from consideration as they did not meet the requirements of our research based on their abstracts. Additionally, 18 papers were further eliminated because they did not address the specific question of this study. Furthermore, 18 studies were excluded due to the unavailability of the full text. Another 7 studies were omitted because they focused on the wrong age group. Lastly, 8 articles were disregarded as they were written in a language other than English. Thus, we based our final analysis on a total of 12 search results that met the criteria for our investigation.

The studies met the standards for inclusion, and we organized the data from these publications in Table 1. 

The search resulted in a total of 44 citations for “microbiota and colorectal cancer”.

The search for “bacteria in colorectal cancer” revealed a total of 72 citations.

The search for “colorectal cancer and biomarkers” resulted in 582 articles.

The search filters we used on PubMed included the following criteria: availability of free full text, randomized controlled trials, English language, adults 18 years or older, from January 2013 to December 2023.

## 3. Results

For this study, we selected 20 studies that were analyzed and included in Table 1. We presented the main conclusions of each study and, using the PICO framework, we answered a clear and focused research question, which is whether CRC is associated with gut microbiota. The main data are presented in Table 1.

### 3.1. Statistical Analysis

The papers included in this meta-analysis provide valuable insights into the connection between microbiota and CRC and, in our opinion, give crucial context for the advancement of future therapies. Descriptive characteristics of study population included in this meta-analysis are included in Table 2 below.

#### 3.1.1. Forest Plot

To analyze the data in Table 1, we have opted for a forest plot as a graphical representation. Meta-analyses and systematic reviews often use this method to present the findings of several research on a certain subject. The purpose of this map is to visually summarize the estimated effect sizes and their confidence intervals from the 12 chosen studies, which allows for an assessment of the general trend and variability in the data.

The forest plot presented showcases the results of this meta-analysis examining the effect sizes of the selected studies on the increase or decrease in biological phyla. Each study’s effect size is represented by a square with larger squares indicating studies that carry more weight in the analysis. The horizontal lines extending from these squares depict the 95% confidence intervals (CI) for each effect size. A positive effect size, shown to the right of the zero line, suggests an increase in biological phyla, while a negative effect size, shown to the left, suggests a decrease. The data indicate that most studies report a positive effect size, with the majority having a *p*-value of 0.00, signifying statistically significant results. Only one study, conducted by Zhong et al. [34], shows a negative effect size, indicating a decrease in biological phyla. This suggests that there is substantial evidence across most studies supporting an increase in biological phyla.

Figure 2 shoes that the overall effect size is 22.38, and this suggests that when all studies are combined, the estimated effect is 22.38. This means that the “INCREASE in biological phyla” in CRC is 22.38% more than the “DECREASE in biological phyla”.

The average CI is from approximately 10 to 35. This means we are 95% confident that the true increase in phyla lies between 10% and 35%.

It can also be seen that when it comes to the weight of studies, larger studies contribute more to the overall effect size.

The overall effect size of 22.38 is a significant and substantial finding, indicating the effectiveness of the intervention of the studies selected for this analysis.

#### 3.1.2. Funnel Plot

Figure 3 shows a funnel plot of the studies which are described in Table 1 using the PICO framework. A funnel plot is typically used to assess publication bias. It is a scatter plot of the effect estimates from individual studies against a measure of each study’s size or precision (e.g., standard error or sample size). Studies with higher precision are plotted toward the top of the graph, closer to the vertical line indicating the estimated overall effect size. The dashed lines represent the 95% pseudo-confidence intervals, creating a funnel shape.

X-axis: Usually represents the effect size (e.g., risk ratio, odds ratio, mean difference). In our case, it corresponds to the effect size. 

Y-axis: Represents a measure of the study’s precision, often the standard error or inverse of the standard error (1/SE).

A symmetrical funnel plot suggests that there is no publication bias. In Figure 3, it can be seen that on each side of the main line, there are 6 studies, which means that our research does not have any publication bias.

Funnel plots are crucial because they help understand whether the results of a meta-analysis might be skewed by the selective publication of studies.

In this plot, the distribution of studies appears somewhat asymmetrical. For instance, the study by Yu et al. [32] is an outlier with high precision but a large effect size, which might indicate a particularly strong effect or possibly a methodological difference. Additionally, studies such as those by Zhong et al. [34], Kordahi et al. [36], and Geng et al. [39] are clustered at the lower precision end, which might indicate smaller sample sizes or higher variability.

Figure 4 shows the changes in the microbiome among the 1363 patients from the 12 studies included in this research. Understanding these changes in bacterial populations has significant implications for CRC diagnosis, prevention, and treatment.

Our research has pinpointed several bacterial species whose presence or abundance in the gut microbiota can be indicative of early-stage CRC.

The comparison between the two sets of taxa underscores a significant shift in the gut microbiome composition in CRC patients. The increased presence of certain bacteria associated with inflammation, infection, and dysbiosis, coupled with the decreased presence of beneficial bacteria, highlights the complex interplay between the microbiome and colorectal cancer. This information is critical for developing microbiome-targeted therapies and preventive strategies, and targeting specific harmful taxa, it may be possible to mitigate the risk or progression of CRC.

### 3.2. Potential Biomarkers for the CRC

The gut microbiota has garnered significant attention as a potential biomarker for colorectal cancer with various bacterial species demonstrating strong associations with the disease. Notably, certain bacterial populations increase in abundance in CRC patients, making them prime candidates for early detection biomarkers. For instance, Yu et al. [32], Thomas et al. [40] and Yachida et al. [41] identified *Fusobacterium nucleatum* consistently in higher concentrations in CRC. This bacterium is known to promote inflammation and tumorigenesis, contributing to the carcinogenic environment in the colon [42]. Similarly, *Bacteroides fragilis*, particularly the enterotoxigenic strains, produce toxins that induce colonic inflammation and have been linked to carcinogenesis [43]. Flemer et al. [30], Kordahi et al. [36], Sheng et al. [38], Geng et al. [39] showed an increase in this bacteria in CRC patients.

*Peptostreptococcus anaerobius* was increased in CRC patients, as shown by Yu et al. [32] and Zeller et al. [33]. Yu et al. [32] further confirmed the same patterns in *Parvimonas micra* and *Solobacterium moorei*, both showing a higher abundance.

These studies underscore the potential of these bacterial species not only as biomarkers for CRC detection but also as targets for therapeutic interventions aimed at modulating the gut microbiota to prevent or manage colorectal cancer.

## 4. Discussion

This extensive metagenomic dataset examined in this study enabled us to analyze the gut microbial virulence believed to contribute to the development of colorectal cancer. 

The link between gut dysbiosis and colorectal cancer involves several complex mechanisms that influence CRC initiation, promotion, and progression [44].

Dysbiosis can lead to chronic inflammation in the gut mucosa, which is characterized by increased levels of pro-inflammatory cytokines and immune cell infiltration [45]. Chronic inflammation creates a microenvironment conducive to carcinogenesis by promoting cell proliferation, inhibiting apoptosis, and stimulating angiogenesis [23,27]. Moreover, dysbiosis-induced inflammation can contribute to the production of reactive oxygen and nitrogen species, causing DNA damage and genomic instability, which are early events in CRC development [46,47].

Our study strengthens the fact that microbial metabolism in the gut plays a critical role in human health and disease, including colorectal carcinogenesis [48]. The gut microbiota metabolizes various dietary components and produces a wide array of metabolites, including short-chain fatty acids (SCFAs) and secondary bile acids, which can impact colorectal carcinogenesis through diverse mechanisms [49,50]. SCFAs, such as acetate, propionate, and butyrate, are produced by the fermentation of dietary fibres and resistant starches by gut bacteria, particularly Firmicutes and Bacteroidetes [51]. Butyrate, in particular, serves as a major energy source for colonic epithelial cells and plays a crucial role in maintaining gut barrier function and modulating immune responses [52]. SCFAs have anti-inflammatory properties and can inhibit the growth of cancer cells by inducing apoptosis, promoting cell cycle arrest, and suppressing tumor cell proliferation [53]. Additionally, SCFAs can regulate gene expression patterns in colonic epithelial cells, modulating various signaling pathways involved in cell growth, differentiation, and apoptosis [51].

Bile acids are synthesized in the liver from cholesterol and secreted into the small intestine, where they aid in the digestion and absorption of dietary fats. Primary bile acids are subsequently metabolized by gut bacteria into secondary bile acids, such as deoxycholic acid (DCA) and lithocholic acid (LCA) [54]. Figure 5 shows the relationship between microbiota and colorectal cancer. It can be seen that the gut microbiota has a crucial role in controlling the onset and advancement of colorectal cancer. The loss of the protective function on the surface of colorectal tumors allows the entry of beneficial microorganisms and their by-products, which subsequently trigger activation.

The results of this study are in line with those of other researchers. For example, Wu et al. showed that at the phylum level, the gut microbiota in healthy controls, adenomas, and CRC was mostly composed of *Firmicutes* and *Bacteroidetes*, with *Proteobacteria*, *Actinobacteria*, *Verrucomicrobia*, *Tenericutes*, and *Fusobacteria* also present [55]. Figure 4 confirms some of these results, as our meta-analysis proved that *Firmicutes*, *Bacteroidetes* and Verrucomicrobia are indeed increasing, but Proteobacteria and Fusobacteria can also decrease in some patients with CRC, while Acinetobacteria mostly decreases. Wirbel et al. [56] also concluded that colorectal carcinogenesis is influenced by *Fusobacterium nucleatum*, *Bacteroides fragilis*, some *Escherichia coli* strains, and intestinal *Clostridium* spp. Our results show similar results, as we found evidence that the above phyla increases in abundance but also decreases in some patients.

One of the most well-documented bacterial species associated with CRC is *Fusobacterium nucleatum*. Research indicates that this bacteria can adhere to and invade colonic epithelial cells, facilitating a pro-inflammatory environment that supports cancer development. For instance, a study by Ou et al. [57] detailed how this bacterium interacts with the immune system, promoting a local inflammatory response that can lead to tumor progression. Moreover, *Escherichia coli* strains, particularly those producing colibactin, are implicated in CRC due to their genotoxic effects, which can induce DNA damage and contribute to carcinogenesis [58]. Studies have shown that *Escherichia coli* can adhere to the intestinal mucosa and release toxins that disrupt cellular processes, further linking these bacteria to CRC [59,60].

Secondary bile acids can disrupt the intestinal epithelial barrier and promote inflammation, further contributing to colorectal carcinogenesis [61]. In addition to SCFAs and secondary bile acids, gut microbiota produce a wide range of other metabolites that can impact colorectal carcinogenesis, including indole derivatives, polyamines, and trimethylamine N-oxide (TMAO) [62]. Indole derivatives, such as indole-3-propionic acid (IPA), have been shown to exhibit anti-inflammatory and anti-tumorigenic effects in the colon by modulating immune responses and inhibiting cell proliferation [63]. Polyamines, such as putrescine and spermidine, are involved in cell growth and proliferation and may contribute to CRC progression when dysregulated [64].

Studies investigating the diagnostic potential of gut microbiota analysis for colorectal cancer have gained significant attention in recent years due to the potential for non-invasive and accurate screening methods.

In our study, we show that Zeller et al. [33] identified microbial signatures associated with CRC and developed a diagnostic model based on microbial biomarkers. The model showed high sensitivity and specificity for detecting CRC, suggesting the potential of gut microbiota analysis for non-invasive CRC screening [33].

In another study, Yu et al. [32] investigated the fecal microbiota composition of CRC patients and healthy controls using 16S rRNA gene sequencing. They identified specific bacterial taxa that were significantly associated with CRC and developed a microbial-based diagnostic classifier. The classifier demonstrated high accuracy in distinguishing CRC patients from healthy individuals, highlighting the potential of gut microbiota analysis for CRC screening. Baxter et al. utilized metagenomic shotgun sequencing to characterize the gut microbiota of CRC patients, adenoma patients, and healthy controls. The researchers identified microbial biomarkers associated with CRC and developed a diagnostic model incorporating these biomarkers. The model accurately differentiated CRC patients from healthy controls and adenoma patients, indicating the potential utility of gut microbiota analysis for CRC detection [32].

Wirbel et al. [65] conducted a large-scale metagenomic analysis of fecal samples from CRC patients and healthy individuals across multiple cohorts. They identified a microbial signature associated with CRC and developed a diagnostic model based on microbial species abundance. The model demonstrated high sensitivity and specificity for CRC detection, suggesting the potential of gut microbiota analysis as a non-invasive screening tool for CRC. Moreover, Zackular et al. [31] investigated the fecal microbiota composition of CRC patients and healthy controls using 16S rRNA gene sequencing. The researchers identified microbial signatures associated with CRC and developed a microbial-based diagnostic test. The test accurately distinguished CRC patients from healthy controls, indicating the potential of gut microbiota analysis for CRC screening [31].

These studies highlight the growing interest in gut microbiota analysis for CRC diagnosis and provide valuable insights into the development of non-invasive screening methods for CRC. Further research and clinical validation are needed to optimize the performance and utility of gut microbiota-based tests in clinical practice.

## 5. Limitations

Studies included in the literature review may vary in terms of study design, sample size, methodology for gut microbiota analysis, and criteria used to define early- and late-onset CRC. This heterogeneity can make it challenging to draw definitive conclusions and may introduce bias or inconsistencies in the results. Different studies may utilize different techniques for gut microbiota analysis, such as 16S rRNA gene sequencing, metagenomic shotgun sequencing, or functional metagenomics. Variability in methodologies and bioinformatics pipelines can lead to differences in the identification and characterization of microbial taxa, potentially affecting the comparability of results across studies.

Numerous confounding factors, such as diet, lifestyle, medication use, comorbidities, and geographical location, can influence the gut microbiota composition in CRC patients. Failure to adequately account for these confounders in the reviewed studies may limit the ability to attribute observed differences in gut microbiota profiles specifically to age of onset of CRC. Also, studies with statistically significant findings may be more likely to be published than those with null or nonsignificant results, leading to publication bias. This bias can skew the overall interpretation of the literature and may overemphasize certain findings while underrepresenting others.

Despite advances in gut microbiota research, our understanding of the functional roles of specific microbial taxa and their interactions with the host and the tumor microenvironment in CRC is still evolving. This incomplete understanding may limit the interpretation of gut microbiota profiles in the context of CRC pathogenesis.

## 6. Conclusions

Conclusions drawn from studies comparing gut microbiota profiles in colorectal cancer can provide insights into the potential role of the gut microbiota in CRC pathogenesis and the differences between these two subtypes of the disease. Studies have reported differences in the gut microbiota composition between early- and late-onset CRC patients. While there is some variability among studies, certain microbial taxa have been consistently associated with CRC regardless of age of onset. However, specific differences in microbial composition in CRC may exist, suggesting potential differences in the underlying pathogenic mechanisms.

Factors such as age-related changes in the gut environment, host immune function, and lifestyle factors may influence the composition and activity of the gut microbiota in CRC patients. These age-related factors could contribute to differences in microbial profiles observed in CRC. Gut microbiota profiles in CRC patients have been linked to tumor molecular subtypes, such as microsatellite instability status and CpG island methylator phenotype. Differences in the prevalence of these molecular subtypes between early- and late-onset CRC may influence the gut microbiota composition and contribute to observed differences.

Understanding the differences in gut microbiota profiles of CRC could have clinical implications for disease management and treatment. Gut microbiota-based biomarkers may aid in CRC risk stratification, early detection, and personalized treatment approaches based on age of onset and microbial profiles.

Further research is needed to elucidate the underlying mechanisms driving differences in microbial composition between these two subtypes of CRC and to explore the clinical implications and therapeutic potential of gut microbiota-based interventions for CRC management.

## Figures and Tables

**Figure 1 ijms-25-07937-f001:**
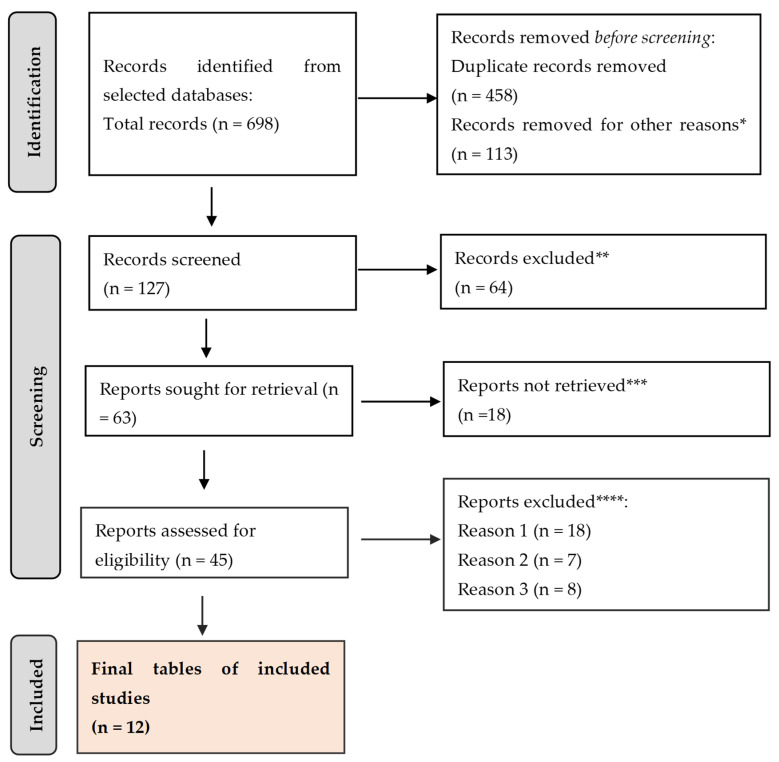
PRISMA framework. * studies are not relevant for the present review. ** studies do not help us to provide an answer to the current research. *** unable to find the full text of the study. **** Reason 1—study on animals/Reason 2—wrong setting/Reason 3—research question not relevant.

**Figure 2 ijms-25-07937-f002:**
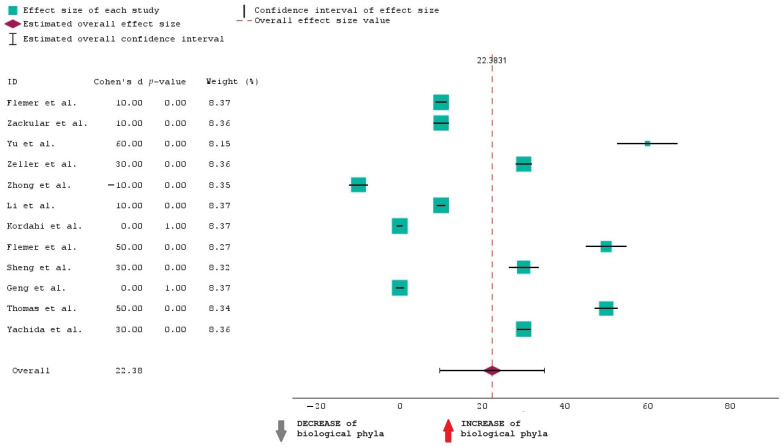
Forest plot of the 12 studies included in the research [30,31,32,33,34,35,36,37,38,39,40,41].

**Figure 3 ijms-25-07937-f003:**
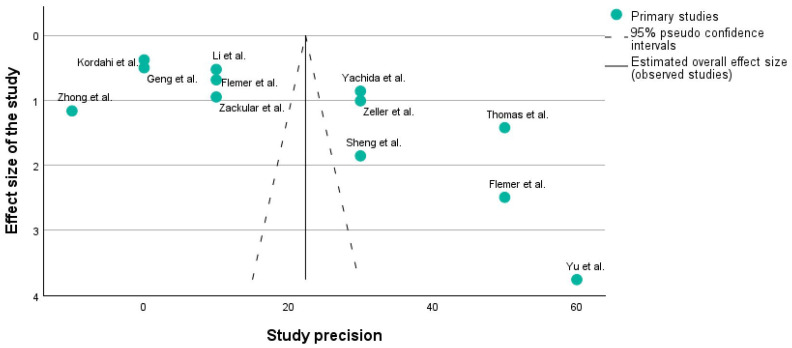
Funnel plot of the 12 studies included in the research [31,32,33,34,35,36,37,38,39,40,41].

**Figure 4 ijms-25-07937-f004:**
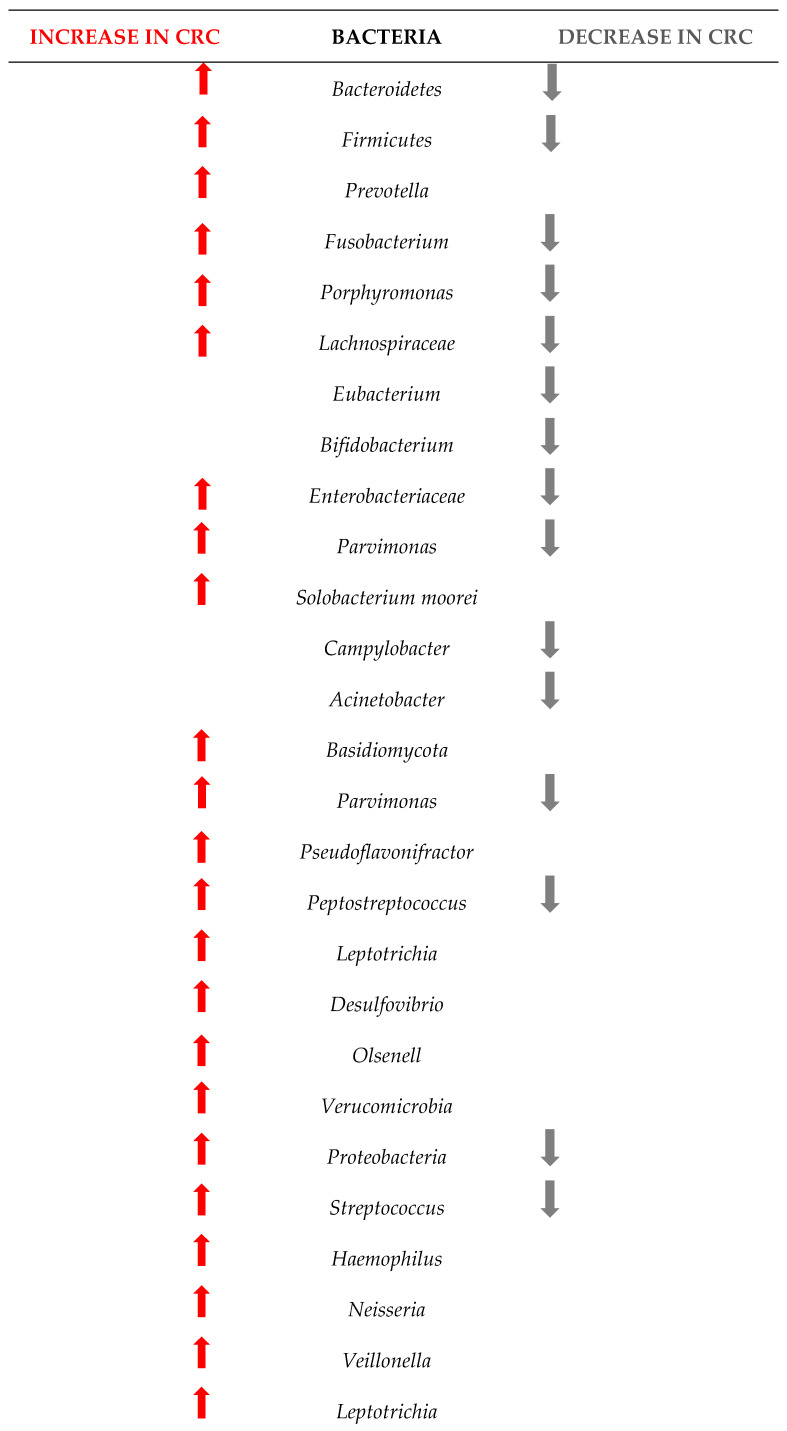
Changes in the bacteria populations among patients with CRC. The red arrow reflets an increase in the bacteria, and the grey arrow a decrease.

**Figure 5 ijms-25-07937-f005:**
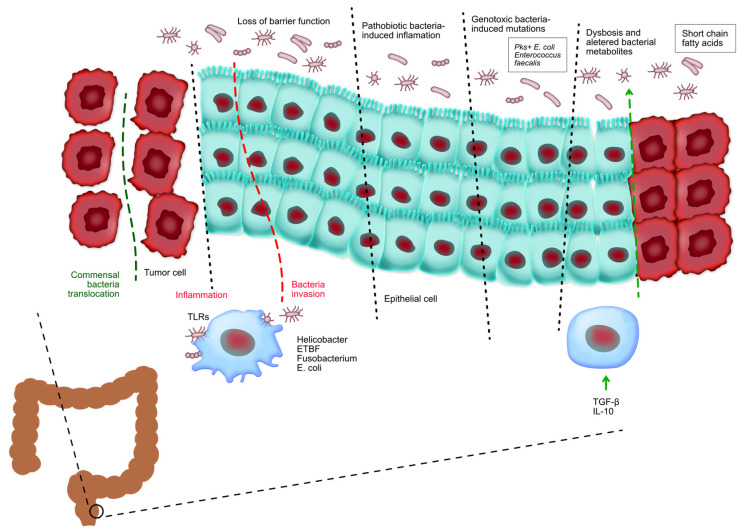
Tumor-associated myeloid cells are stimulated by the presence of tumors and contribute to the promotion of inflammation that supports tumor growth. Pathogenic bacteria infiltrate healthy colorectal tissues and facilitate the initiation of inflammation and cancer. Bacteria harboring genotoxic markers facilitate the accumulation of genetic abnormalities in the cells lining the intestines, hence initiating the development of CRC. Chronic inflammation in inflammatory bowel disease and CRC changes the makeup of beneficial bacteria and leads to an imbalance known as dysbiosis, which in turn worsens tissue inflammation. The development of colorectal cancer is regulated by food metabolites produced by commensal bacteria. These metabolites affect the growth and survival of tumor cells as well as the ability of immune cells to eliminate tumors.

**Table 1 ijms-25-07937-t001:** Results of clinical studies investigating the manifestations of gut microbiota in relation with CRC.

Study	PICO Framework	Key Results	Microbiota Involvement in CRC
Flemer et al. [30]	*Population:* total of 136 participants *Intervention:* microbiota composition was analyzed using 16S rRNA amplicon sequencing, while the expression of host genes related to colorectal cancer development and immune response was measured using real-time quantitative PCR *Comparison:* 59 patients undergoing surgery for CRC, 21 individuals with polyps and 56 healthy controls *Outcome:* to assess if the microbiome profiles associated with CRC vary from those in healthy individuals and if they are correlated with certain gene-expression patterns in the mucosa	The microbiota of patients with CRC showed differences compared to the microbiota of healthy individuals. However, these modifications were not limited to the malignant tissue alone. The study identified differences between distal and proximal malignancies and found that the composition of the fecal microbiota only partly matched that of the mucosal microbiome in CRC. Patients with colorectal cancer may be categorized based on higher-level structures of mucosal-associated bacterial co-abundance groups (CAGs) that reflect the enterotypes. The correlation between CRC-associated CAGs and the expression of host immunoinflammatory response genes varied. The presence of *Firmicutes* Cluster 1 and *Bacteroidetes* Cluster 1 in the microbiota of CRC patients was significantly lower. On the other hand, the abundance of *Firmicutes* Cluster 2, *Prevotella* Cluster, Pathogen Cluster, and *Bacteroidetes* Cluster 2 was higher in the CRC biopsy microbiota.	INCREASE in: ▪ *Bacteroidetes* ▪ *Firmicutes* ▪ *Prevotella* and DECREASE in: ▪ *Bacteroidetes* ▪ *Firmicutes*
Zackular et al. [31]	*Population:* total of 90 participants *Intervention:* microbiota was analyzed by sequencing the V4 region of the 16S rRNA gene from the feces of each individual using the Illumina MiSeq sequencing platform *Comparison:* 30 patients with CRC, 30 patients with adenoma, and 30 healthy individuals as control group *Outcome:* to compare the microbiome of healthy individuals, persons with adenomas, and patients with colorectal carcinomas	Compared to healthy subjects, individuals with carcinomas were found to have higher levels of certain microorganisms, specifically *Fusobacterium*, *Porphyromonas*, *Lachnospiraceae* (OTUs 31, 59, 32, 116, 85), and *Enterobacteriaceae.* On the other hand, they had lower levels of microorganisms associated with *Bacteroides*, *Lachnospiraceae* (OTUs 23, 30, 253, 136), and *Clostridiales*.	INCREASE in: ▪ *Fusobacterium* ▪ *Porphyromonas* ▪ *Lachnospiraceae* ▪ *Enterobacteriaceae* and DECREASE in: ▪ *Bacteroides* ▪ *Lachnospiraceae* ▪ *Clostridiales*
Yu et al. [32]	*Population:* even though the study collects data from multiple countries, we selected only the ones in which feces were analyzed. Thus, for this review, we selected the 128 individuals from China *Intervention:* metagenomic sequencing on the stool samples *Comparison:* 74 patients with CRC and 54 control subjects *Outcome:* to examine taxonomic differences between CRC-associated and control microbiomes, and to identify microbial taxa contributing to the dysbiosis	The findings indicate a disbyosis in the gut microbiota of people with CRC. Leucine promotes both the creation and breakdown of proteins, indicating potential connections between leucine metabolism and cancer. Significant relationships between illness state and various KEGG orthologous groupings were observed at the gene level. The presence of Eubacterium ventriosum was consistently higher in the microbiomes of the control group. However, *Parvimonas micra*, *Solobacterium moorei*, *Fusobacterium nucleatum*, and *Peptostreptococcus stomatis* were consistently found in higher abundance in the microbiomes of patients with colorectal cancer. At the phylum level, only the *Fusobacteria* and *Basidiomycota* were shown to be considerably more abundant in microbiomes linked with colorectal cancer.	INCREASE in: ▪ *Parvimonas micra* ▪ *Solobacterium moorei* ▪ *Fusobacterium nucleatum* ▪ *Peptostreptococcus stomatis* ▪ *Fusobacterium* ▪ *Basidiomycota*
Zeller et al. [33]	*Population:* 391 participants from France, Germany, Denmark and Spain *Intervention:* metagenomic sequencing of fecal samples to identify taxonomic markers that distinguished CRC patients from tumor-free controls *Comparison:* 358 healthy individuals, 42 patients with adenoma, 91 patients with CRC *Outcome:* to compare the microbiota from patients with adenomas to neoplasia-free controls and to patients with colorectal cancer	Despite variations in patient country or origin, experimental methodologies, and analytic methodology, there are similarities in the relative abundances of certain species from fecal samples from CRC patients. These findings indicate that analyzing fecal samples may provide insights into the metabolic and functional capabilities of the colonic microbiome in the tumor setting.	INCREASE in: *Fusobacterium* *Pseudoflavonifractor* *Peptostreptococcus* *Leptotrichia* *Porphyromonas* *Desulfovibrio* *Parvimonas* *Olsenell* and DECREASE in: *Eubacterium* *Ruminococcus* *Bifidobacterium* *Campylobacter* *Acinetobacter*
Zhong et al. [34]	*Population:* 40 participants *Intervention:* 16S rRNA gene sequencing of the microbiota present in the normal colorectal mucosa and colorectal polyps *Comparison:* microbial composition of the colorectal polyp tissue and fecal samples (24 adults) and of the normal intestinal mucosal tissue and fecal samples (16 adults) *Outcome:* to find out if the microbial structure in feces differs from that in tissues of polyp and normal mucosa	The differential genera seen in the normal colorectal mucosa and the stool sample of individuals with colorectal polyps were similar to those observed between the fecal sample and the polyp tissue. In both groups, the levels of *Bacteroides*, *Prevotella-2*, and *Agathobacter* were higher, while the levels of *Haemophilus*, *Escherichia_Shigella*, *Fusobacterium*, and *Streptococcus* were lower in feces compared to the normal mucosa in both groups or polyp tissues. There was no notable difference in the presence of *Fusobacterium* between the typical colorectal mucosa and polyps in individuals with colorectal polyps. However, it was much greater in both the mucosa and polyps compared to the fecal matter. There was a greater presence of *Fusobacterium* in the normal intestinal mucosa of healthy persons compared to the polyp group.	INCREASE in: *Firmicutes* *Verucomicrobia* and DECREASE in: *Proteobacteria* *Fusobacteria* *Acinetobacteria*
Li et al. [35]	*Population:* 98 CRC participants *Intervention:* microbial community genomic DNA was extracted from 200 mg fecal samples or 300 mg tissue samples *Comparison:* three types of colorectal mucosa (tumor mucosa, para-cancerous mucosa, normal mucosa) and feces *Outcome:* to find out if there are variations between tumor mucosal microbiota and normal mucosal microbiota	The study found important variations between tumor mucosal microbiota and normal mucosal microbiota, but no notable difference was seen in the microbiota between the tumor and para-tumor mucosa, as well as between the para-tumor and normal mucosa. This suggests that the microbiota in the para-cancerous mucosa serves as an intermediate stage between the microbiota in the tumor and normal mucosa. The significant changes in the composition of the fecal microbiota, when compared to the mucosal microbiota, suggest that utilizing the fecal microbiota as a definition for the mucosal microbiota carries a certain level of risk. A robust link was identified between a positive result on the fecal occult blood test and the presence of *Fusobacterium*. This suggests that this particular genus, known for its ability to attach to and invade the intestines, is strongly associated with intestinal bleeding. In addition, we have discovered six main types of bacteria, namely *Fusobacterium*, *Gemella*, *Campylobacter*, *Peptostreptococcus*, *Alloprevotella*, and *Parvimonas*, that consistently show higher presence in tumor tissue compared to normal tissue and/or in tissue compared to feces.	INCREASE in: *Firmicutes* *Proteobacteria* *Fusobacteria (tumour mucosa)* and DECREASE in: *Fusobacteria (feces)* *Acinetobacteria*
Kordahi et al. [36]	*Population:* 40 participants *Intervention:* two biopsies were collected from CRC patients: one from the area surrounding the polyp or polyp adjacent and another from the macroscopically healthy mucosa situated at least 10 cm distant from the polyp, and one biopsy from the healthy mucosa of control subjects *Comparison:* 31 patients with CRC and 9 individuals in the control group *Outcome:* to gain insight into the microbial microenvironment of the pre-neoplastic polyp as compared to the healthy mucosa	There is a clear difference in the types of microorganisms present in the mucosa that is next to pre-neoplastic colonic polyps compared to the mucosa of individuals who are polyp-free. During the first phases of polyp development, it is possible that an imbalanced and inflammatory gut environment enables the growth or establishment of nontoxigenic B. fragilis that possess an abundance of genes responsible for the production of lipopolysaccharides.	INCREASE in: *Firmicutes* *Bacteroidetes* and DECREASE in: *Proteobacteria* *Acinetobacteria*
Flemer et al. [37]	*Population:* 234 participants *Intervention:* the microbiota was analyzed from oral swabs, colonic mucosae and stool of the subjects *Comparison:* 99 CRC patients, 32 colorectal polyps patients and 103 Controls. *Outcome:* to find out if microbiota alterations are linked with colorectal cancer	The most prevalent bacterial species found in all oral swab samples were *Streptococcus* (over 30%), *Haemophilus* (14.2%), *Neisseria* (almost 9%), Prevotella (6.6%), *Fusobacterium* and *Veillonella* (over 5%), *Leptotrichia* and *Rothia* (almost 4%), *Actinomyces* (2.9%), and *Porphyromonas* (2.4%).	INCREASE in: *Streptococcus* *Haemophilus* *Neisseria* *Prevotella* *Veillonella* *Leptotrichia* *Rothia* *Actinomyces* and DECREASE in: *Porphyromonas* *Parvimonas* *Fusobacterium*
Sheng et al. [38]	*Population:* 66 CRC patients *Intervention:* all paired tumor and adjacent normal tissue samples were dissected and frozen immediately after collection and stored at −80 °C until DNA extraction *Comparison:* microbiota of the tumor mucosa with that one of the normal mucosa *Outcome:* to find out if there are any microbiota linked with colorectal cancer	7 microbe genus (*Fusobacterium*, *Faecalibacterium*, *Akkermansia*, *Ruminococcus*, *Parabacteroides*, *Streptococcus*, and *Ruminococcaceae*) were significantly different between tumor and adjacent normal tissues; and 5 microbe genus (*Bacteroides*, *Fusobacterium*, *Faecalibacterium*, *Parabacteroides*, and *Ruminococcus*) were also different between distal and proximal CRC segments.	INCREASE in: *Bacteroides* *Escherichia Shigella* *Prevotella* *Faecalibacterium* *Akkermansia* *Ruminococcus* *Parabacteroides* Clostridium XVIII *Lachnospiraceae* *Ruminococcaceae* and DECREASE in: *Fusobacterium* *Enterobacteriaceae* *Peptostreptococcus* *Parvimonas* *Streptococcus* *Gemella* *Acinetobacter*
Geng et al. [39]	*Population:* 8 patients with CRC *Intervention:* 16S rRNA gene sequences of biopsy samples taken from both healthy and tumor mucosa *Comparison:* between microbiota of normal mucosa and tumor mucosa *Outcome:* to quantitatively evaluate the differences of bacterial communities and compositions between eight tumor/normal pairs	The research identified two distinct patterns of change associated with each of the three major bacterial taxa in the gut (*Roseburia*, *Microbacterium*, and *Anoxybacillus*).	INCREASE in: *Roseburia* *Bacteroidetes* and DECREASE in: *Micobacterium* *Anoxybacillus*
Thomas et al. [40]	*Population:* 764 participants *Intervention:* participants underwent colonoscopy to diagnose CRC, adenoma, or to confirm the absence of disease, with samples collected before diagnosis or beginning of treatment Comparison: stool microbiome from 313 CRC patients, 143 subjects with adenoma and 303 controls *Outcome:* to establish consistent correlations between the gut microbiota and colorectal cancer	There is a much greater number of species that are more abundant in individuals with colorectal cancer compared to those without the disease (controls). The functional capacity of the microbiome was shown to be strongly linked to colorectal cancer samples in comparison to healthy subjects.	INCREASE in: *Fusobacterium nucleatum* *S. moorei* *Porphyromonas asaccharolytica* *Parvimonas micra* *Peptostreptococcus stomatis* *Clostridium symbiosum* *Gemella morbillorum* *Streptococcus gallolyticus* and DECREASE in: *Gordonibacter pamelae* *Bifidobacterium catenulatum*
Yachida et al. [41]	*Population:* 616 patients *Intervention:* metagenomic and metabolomic markers were identified to separate cases of intramucosal carcinoma from the healthy controls *Comparison:* 146 healthy subjects and 470 patients with different stages of CRC *Outcome:* to demonstrate possible associations between the gut microbiota and colorectal cancer	The abundance of certain species belonging to the phyla *Firmicutes*, *Fusobacteria*, and *Bacteroidetes* was consistently higher and correlated with the severity of malignancy. *Fusobacteria* of higher taxonomic rank were seen in at least two distinct phases, but numerous species from the *Firmicutes* and *Bacteroidetes* phyla were exclusive to a particular stage. Within the phylum *Proteobacteria*, a significant number of species were shown to be increased in patients with colorectal cancer. The population of the *Bifidobacterium* genus was mostly reduced in early stages of CRC.	INCREASE in: *Firmicutes* *Fusobacterium nucleatum* *Atopobium parvulum* *Actinomyces odontolyticus* and DECREASE in: *Bifidobacterium*

**Table 2 ijms-25-07937-t002:** Population characteristics included in the selected studies.

Study	CRC Mucosa or Stool Samples (*n* = 1363)	Other Polyps or Adenoma Mucosa or Stool Samples (*n* = 268)	Healthy Mucosa or Stool Samples (*n* = 1252)
Flemer et al. [30]	59	21	56
Zackular et al. [31]	30	30	30
Yu et al. [32]	74	N/A	54
Zeller et al. [33]	91	42	358
Zhong et al. [34]	24	N/A	16
Li et al. [35]	98	N/A	98
Kordahi et al. [36]	31	N/A	9
Flemer et al. [37]	99	32	103
Sheng et al. [38]	66	N/A	66
Geng et al. [39]	8	N/A	8
Thomas et al. [40]	313	143	308
Yachida et al. [41]	470	N/A	146

## Data Availability

Not applicable.

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
