# Peer review of "Gut Microbiota Signatures in Colorectal Cancer as a Potential Diagnostic Biomarker in the Future: A Systematic Review"

_ijms, 2024, doi:10.3390/ijms25147937_

Round 1

Reviewer 1 Report

Comments and Suggestions for Authors

1.- In Table 1, regarding the key results from Flemer et al., please correct the phrase "showed distinct differences." Additionally, under microbiota involvement, there is a discrepancy where Bacteroidetes and Firmicutes are indicated to both increase and decrease. Could you verify or clarify this point?

2.- In Table 1, for Zackular et al., it is noted that Lachnospiraceae both increased and decreased. It is necessary to specify the taxonomic level at which these changes were observed for each study reviewed.

3.- Line 206: The phrase “statistical analysis” would be more precise without the addition of "of results."

4.- Line 218: It is essential to establish a dedicated results section here. Furthermore, the subsequent lines require rephrasing for enhanced readability. The text currently contains numerous isolated sentences. The results should be articulated in clear, coherent paragraphs.

5.- Line 249: The reference to "table 3" seems more akin to a figure. The text is unclear, and the lettering on the images appears scrambled, making it difficult to discern their origin. Are these histological images from this study? The recruitment of patients has not been declared. Additionally, the genus described in this "table" is ambiguous with repeated letters. Consider either eliminating this table/figure or revising it for clarity.

6.- Lines 259 to 269: These paragraphs function more as a discussion. Consider relocating this content accordingly.

7.- In the discussion section, aim to consolidate comments and information related to each idea into coherent paragraphs, avoiding single-sentence paragraphs.

8.- Please revise the entire manuscript to enhance readability and eliminate typographical errors that are present throughout the document.

Reviewer 2 Report

Comments and Suggestions for Authors

The subject of this review is interesting, since the clear identification of microbial species that potentially act as indicators of CRC risk can become a useful tool. The article is systematic in its development and it is easy to follow the sequence of work followed by the authors. However, it is somewhat ambiguous when it comes to establishing the identity of the species that can act as markers. In this sense, the work is excessively descriptive, without a clear conclusion in terms of identification of specific microorganisms. I understand that it is not easy, given that the results are sometimes contradictory, but perhaps it would be advisable to lower the expectations derived from the title of the article.

Two comments regarding microbial taxonomy. On the one hand, the use of italics when naming genera and species and, on the other hand, the updating of the names of bacterial phyla. In 2021 there was a revision in relation to this issue, whereby the names that appear in the text are no longer valid.

An additional comment: reference 46 does not appear in the text.

Round 2

Reviewer 1 Report

Comments and Suggestions for Authors

All the previous commnets were addresed in this version.